# A Fast, Efficient, and Tissue-Culture-Independent Genetic Transformation Method for *Panax notoginseng* and *Lilium regale*

**DOI:** 10.3390/plants13172509

**Published:** 2024-09-06

**Authors:** Jie Deng, Wenyun Li, Xiaomin Li, Diqiu Liu, Guanze Liu

**Affiliations:** 1Faculty of Life Science and Technology, Kunming University of Science and Technology, Kunming 650500, China; dj2826122825@126.com (J.D.); liwenyun97@163.com (W.L.); lxmnlb@163.com (X.L.); 2State Key Laboratory of Conservation and Utilization of Bio-Resources in Yunnan, The Key Laboratory of Medicinal Plant Biology of Yunnan Province, National & Local Joint Engineering Research Center on Germplasms Innovation & Utilization of Chinese Medicinal Materials in Southwest China, Yunnan Agricultural University, Kunming 650201, China; 3Yunnan Seed Laboratory, Yunnan Agricultural University, Kunming 650201, China

**Keywords:** *Agrobacterium*-mediated genetic transformation, meristem injection, *Panax notoginseng*, *Lilium regale*

## Abstract

The *Agrobacterium*-based transgenic technique is commonly used for gene function validation and molecular breeding. However, it is not suitable for plants with a low regeneration capacity or a low transformation rate, such as *Panax notoginseng* (Burk) F.H. Chen and *Lilium regale* Wilson. In this study, a novel *Agrobacterium* transformation method based on injection in the meristems was developed using *P. notoginseng* and *L. regale* as experimental models. PCR analysis confirmed the successful integration of the reporter gene *DsRed2* (*Discosoma striata red fluorescence protein 2*) into the genome of two experimental models. QRT-PCR and Western blot analysis demonstrated the transcriptional and translational expression of DsRed2. Additionally, laser confocal microscopy confirmed the significant accumulation of the red fluorescent protein in the leaves, stems, and roots of transformed *P. notoginseng* and *L. regale*. Most importantly, in the second year after injection, the specific bright orange fluorescence from DsRed2 expression was observed in the transgenic *P. notoginseng* and *L. regale* plants. This study establishes a fast, efficient, and tissue-culture-independent transgenic technique suitable for plants with a low regeneration capacity or a low transformation rate. This technique may improve the functional genomics of important medicinal and ornamental plants such as *P. notoginseng* and *L. regale*, as well as their molecular breeding.

## 1. Introduction

The *Agrobacterium*-based transgenic technique has been extensively used for gene function validation and genetic modification due to its practicality, cost-effectiveness, and high efficiency [1]. The key steps of this technique include explant preparation, explant infection, co-cultivation with *Agrobacterium*, explant regeneration, and the screening of transgenic lines [2], but tissue culture is a limiting step for some species that exhibit a low regenerative capacity and low rate of genetic transformation [3]. Therefore, the *Agrobacterium*-based transformation method is particularly difficult to apply to plants such as *Panax* spp. and *Lilium* spp. [4]. For example, *Panax notoginseng* (Burk) F.H. Chen and *Lilium regale* Wilson are both perennial plants with important economic value, but these two species have a low tissue regeneration ability and transformation rate.

The plants of *Panax* spp. are well-known medicinal herbs with valuable medicinal properties, primarily attributed to their saponin content [5]. However, the lack of a well-established genetic transformation system has hindered further exploration of the saponin biosynthesis pathways and the development of elite *P. notoginseng* cultivars with high and consistent yields and quality [6]. Adventitious roots were regenerated from *Panax ginseng* roots through tissue culture and transgenic *P. ginseng* roots were obtained via *Agrobacterium*-based transformation [7]. RNA interference of *PgRg1-3* in the *P. ginseng* roots was mediated by *Agrobacterium rhizogenes* C58C1 [8]. *PgERF120* was overexpressed in the hairy roots of *P. ginseng* roots based on *Agrobacterium tumefaciens* A4 transformation [9]. *P. notoginseng* cell lines overexpressing *Panax japonicus βAS* (*β-amyrin synthase*) were obtained via *Agrobacterium*-mediated genetic transformation in order to reveal the influence of *PjβAS* on saponin biosynthesis in *P. notoginseng* [10]. The current transformation method involves traditional *Agrobacterium*-based techniques applied to callus tissue induced from *P. notoginseng* explants, but successful generation of transgenic *P. notoginseng* plants has not been achieved using this approach [11]. At the same time, aseptic operation makes the acquisition of genetically modified materials more time-consuming and more difficult.

Similarly, *Lilium* spp., known for their wide range of flower shapes, fragrances, and colors, hold significant ornamental value and rank as the sixth-largest fresh cut flower globally [12]. However, the traditional *Agrobacterium*-mediated transformation method, characterized by its low efficiency, lengthy cycle, and demanding sterile operation, is not widely applicable in molecular breeding efforts for lilies [13]. Although the transformation rate of *Lilium* ‘Sorbonne’ was improved by adjusting the composition of the culture medium; only 14 positive transgenic *Lilium* ‘Sorbonne’ lines out of 173 lines were successfully identified [14]. Fu and co-authors [15] found that the transformation rate of *Lilium brownie* embryogenic callus was 65.56%. The β-glucuronidase (GUS) gene and green fluorescent protein (GFP) gene were successfully expressed in *L. regale* via pollen magnetofection, but a complete transgenic plant was not produced [16]. Therefore, the transformation rate of lilies is still low, and the protocols remain difficult and complex. *L. regale*, a wild lily species with high resistance to various stresses, remains underexplored due to the lack of a mature transformation system, hindering the elucidation of its resistance mechanisms and the utilization of its resistance-related genes to improve lily disease resistance. Establishing an efficient genetic transformation system is crucial for enhancing the ornamental and disease-resistant traits of lilies to improve the economic value of *Lilium* spp.

Establishing a fast, efficient, and tissue-culture-independent genetic transformation method is important for the fundamental research and molecular breeding of *Panax* spp. and *Lilium* spp. However, the transitional *Agrobacterium*-based transformation method is not suitable for some perennial plants. To address these challenges, we utilized *P. notoginseng* and *L. regale* as two experimental models to establish a rapid, efficient, and tissue-culture-independent genetic transformation system. Every spring, the aboveground part of *P. notoginseng* and *L. regale* grows up depending on the meristems located in the rhizome and bulb, respectively. Therefore, the meristems in the *P. notoginseng* rhizome and *L. regale* bulb could be excellent targets for genetic transformation. Additionally, DsRed2 (*Discosoma striata* red fluorescence protein *2*) is a red fluorescent protein isolated from *Discosoma* sp. that was found to be a better reporter gene than GUS and GFP because its excitation light was higher than that of GUS and GFP; thus, it can avoid plant autofluorescence [17]. DsRed2 has been used to screen transgenic cotton (*Gossypium herbaceum*) and transgenic peanut (*Arachis hypogaea*) via fluorescence microscopy and handheld fluorescence observation equipment through the detection of red and bright orange fluorescence, respectively [17,18]. Therefore, *Agrobacterium tumefaciens* of the LBA4404 strain containing the vector pCAMBIA2300::35S::DsRed2 were injected into the meristems of *P. notoginseng* rhizome and *L. regale* bulb. Throughout the 40–60 days of cultivation after *Agrobacterium* injection, a series of experiments were conducted to evaluate the expression levels and activity of the reporter gene in transgenic plants.

## 2. Results

### 2.1. Establishment of Agrobacterium-Based Injection Rhizome Transformation Method in P. notoginseng

*P. notoginseng* is a well-known Chinese traditional herbal medicine plant, but its functional genomics and genetic breeding features, including the improvement of disease resistance and active compound biosynthesis, have been limited by the lack of a stable genetic transformation system. Notably, *P. notoginseng* is a perennial plant with meristems in the rhizome, where new plants are produced every spring. In this study, we developed a genetic transformation method for *P. notoginseng* by injecting *A. tumefaciens* LBA4404 containing *DsRed2* into the rhizome (Figure 1). All the tested *P. notoginseng* plants sprouted on the rhizome after 30–40 days of injection, and the *P. notoginseng* plants showed a similar phenotype as the wild-type (WT) plants (Figure 2A).

The leaves from 71 plants derived from injected rhizomes were used to extract the genomic DNA, and the wild-type *P. notoginseng* was included as the negative control in the PCR analysis and subsequent experiments. Gel electrophoresis of the PCR products revealed that 61 plants contained the *DsRed2* gene in their genomes (Figure 2B and Appendix A). The transformation rate was approximately 85.91%. Furthermore, qRT-PCR analysis confirmed the accumulation of a large number of *DsRed2* transcripts in both the leaves and stems of seven positive transgenic *P. notoginseng* plants (Figure 2C,D), with PT-4 and PT-8 showing the highest expression levels in the stems and leaves, respectively.

Moreover, Western blot analysis further confirmed the expression of the red fluorescent protein in the leaves of all tested *DsRed2* transgenic *P. notoginseng* plants, including PT-1/3/4/5/8/11/13 (Figure 2E). Additionally, red fluorescence was detected in the leaves, stems, and roots of the *DsRed2* transgenic *P. notoginseng* plant (PT-3) using a laser scanning confocal microscope, but no specific red fluorescence was found in the WT plant, indicating that the red fluorescent protein expressed by *DsRed2* in transgenic *P. notoginseng* possesses normal biological activity (Figure 2F). Furthermore, in the second year after injection, bright orange fluorescence was detected in the rhizome and leaves of *DsRed2* transgenic *P. notoginseng*, but this specific fluorescence was not present in the WT plant (Figure 2G). These results demonstrate the successful integration of *DsRed2* into the genome of *P. notoginseng* and the stable expression achieved through the injection of *A. tumefaciens* into the *P. notoginseng* rhizome. This method only takes 50–60 days from transgenic operation to the acquisition of positive transgenic plants, and its use means that complicated aseptic operations and tissue culture can be avoided.

### 2.2. Agrobacterium-Based Injection in the Bulb Meristems Is a Fast and Efficient L. regale Genetic Transformation Method

Lilies are globally important fresh-cut flowers, but explorations of the molecular biology of lilies and breeding research have been hindered by the low efficiency of stable genetic transformation. Lily bulbs have a high reproduction ability due to the presence of the meristems in the innermost bulb layer. To explore a more efficient method for stable genetic transformation in lilies, in this study, we injected *A. tumefaciens* carrying *DsRed2* into the bulb meristems of *L. regale* plants (Figure 3). All *L. regale* bulbs sprouted new plants from the meristems within 25–30 days after injection, and the *L. regale* plants from injected bulbs showed the same phenotype as the WT plant (Figure 4A). Genomic DNA was extracted from the leaves of *L. regale* plants, and PCR analysis indicated that 71 out of 82 tested plants showed the same specific DNA fragments as the positive control (plasmids of pCAMBIA2300::35S::DsRed2) (Figure 4B and Appendix A). The transformation rate was approximately 86.58%. Moreover, a high expression level of *DsRed2* was detected in the leaves of positive transgenic *L. regale* plants, with plants T-5/6/19/21 showing higher expression levels compared to other plants (Figure 4C).

Additionally, Western blot analysis with the DsRed2 antibody revealed that the accumulation of the DsRed2 protein in the leaves of positive transgenic *L. regale* plants (T-6/19/21) was higher than that in the leaves of T-1/5/20. On the contrary, DsRed2 was not found in the leaves of the WT *L. regale* upon Western blot analysis (Figure 4D). Furthermore, observation with a laser scanning confocal microscope confirmed the presence of red fluorescence in the leaves, stems, roots, and scales of transgenic *L. regale* plants (T-6), while no specific red fluorescence was observed in the WT *L. regale* (Figure 4E). Using a handheld fluorescence detector, we found that bright orange fluorescence specifically accumulated in the bulb and shoots of *DsRed2* transgenic *L. regale* plants in the second year after injection (Figure 4F). Generally, through using this method, positive transgenic *L. regale* can be obtained after 40–50 days. These results, obtained using a method involving *Agrobacterium*-based injection in the bulb meristems, which proved to be a rapid and efficient approach suitable for the genetic transformation of *L. regale*, demonstrate the successful integration and stable expression of DsRed2 in transgenic *L. regale* plants.

## 3. Discussion

Functional genomics and molecular breeding studies require efficient and rapid genetic transformation systems. However, traditional *Agrobacterium*-based transformation methods and gene gun techniques have some limitations [19]. The traditional *Agrobacterium*-based method is characterized by a long transformation cycle and low efficiency, and it is unsuitable for plants with limited regeneration capabilities [20]. On the other hand, the gene gun method relies on plants’ regenerative abilities and requires expensive equipment [19]. These methods are not ideal for plants like *P. notoginseng*, which have limited regeneration abilities [21]. Additionally, traditional genetic transformation methods require stringent sterile operations [22], which can significantly reduce plant survival, regeneration, and transformation rates, thereby limiting their application in plants like lilies [3].

For *Panax* spp., there is currently no research indicating that transgenic plants can be successfully developed through the traditional genetic transformation method. It is important to note that the transformation method established in this study showed a transformation efficiency of up to 85.7% in *P. notoginseng*, only taking 50–60 days to achieve this. For *Lilium* spp., it takes at least 90–120 days to obtain the conventional transgenic plants, and the transformation rate is on average 65.6% [13]. In this study, it only took 40–50 days from the injection of *Agrobacterium* to the acquisition of transgenic *L. regale*, and we avoided the complex steps of aseptic operation and tissue culture. At the same time, its efficiency is higher than that of the traditional *Agrobacterium*-based transformation method, reaching 86.58%, making it a more suitable transformation method in terms of applicability to *Lilium* spp.

Recently, Cao et al. [23] established a novel *Agrobacterium*-based transformation method named cut–dip–budding that does not need sterile operations and developed transgenic herbaceous and tuberous plants. Moreover, this method was also used to generate transgenic succulent plants [24]. The injection of *A. tumefaciens* into the meristems in stems successfully generated some stable transgenic plants, such as the sweet potato (*Ipomoea batatas*), potato (*Solanum tuberosum*), and bayhops (*Ipomoea pes-caprae*) [25]. The above two methods are practicable in the plants which have high redifferentiation capabilities, but the transformation methods for plants with low differentiation abilities need further exploration. In this study, we introduced a fast and efficient *Agrobacterium* transformation method based on the injection of *A. tumefaciens* LBA4404 containing *DsRed2* into the meristems of the *L. regale* bulb and *P. notoginseng* rhizome. This method eliminates the need for sterile operations and meets the requirements for plants with low regeneration abilities. The successful expression of the reporter gene *DsRed2* was demonstrated in multiple organs of the *P. notoginseng* and *L. regale* plants (Figure 2 and Figure 4). Protein expression of DsRed2 was confirmed via Western blot analysis, and the presence of red fluorescence in multiple organs of positive transgenic *P. notoginseng* and *L. regale* plants indicated normal activity of the red fluorescent protein (Figure 2E,F and Figure 4D,E).

Most importantly, in the second year after injection, specifically, bright orange fluorescence was observed in the transgenic *P. notoginseng* and *L. regale* plants (Figure 2G and Figure 4F), evidencing the stable expression of *DsRed2*. This study provides robust technical support for fundamental research on plants with low transformation rates and low regeneration abilities. In addition, as perennial plants, it takes 2–3 and 6–8 years for *P. notoginseng* and *L. regale* to generate seeds, respectively; therefore, the transgenic plants featured in this study are the chimera plants. Nonetheless, the genetic features of *DsRed2* require further investigation in order to clearly understand the stability of the transgenic method established in this study.

Furthermore, the gene editing technique based on CRISPR/Cas has been extensively used for plant genome editing to improve important traits without the need for inserting exogenous genes [26]. However, CRISPR/Cas-mediated genome editing relies on a gene delivery system, such as the *Agrobacterium*-based transformation method [27]. Therefore, the development of a rapid and efficient transformation method will facilitate the application of CRISPR/Cas technology in a variety of plant species. There have been no reports on gene editing in *P. notoginseng*, while there are two studies reporting on the use of a CRISPR/Cas9-based system in relation to lilies, with these being published in 2019 and 2020, respectively [13,28]. However, the low transformation rate remains a serious issue affecting the efficiency of lily CRISPR/Cas9 technology. With its high efficiency, this *Agrobacterium* transformation method, based on meristem injection, may bring hope for the application of CRISPR/Cas technology in *P. notoginseng* and *L. regale*.

## 4. Materials and Methods

### 4.1. Plant Materials

After collection from three-year-old plants, the coat of *P. notoginseng* seeds were removed and sown in damp sands with 20–30% humidity before being placed in a refrigerator (8 °C) for low-temperature treatment. After 30 days, the seeds were sown in a shed with a sunshade net. In addition, the *L. regale* seeds were sown in nutrient soil (vermiculite/perlite/nutrient soil = 2:1:1) and cultivated in a greenhouse (25 °C, natural light). One-year-old *P. notoginseng* and *L. regale* plants were used for gene transformation.

### 4.2. A. tumefaciens Strain, Binary Vector, and Culture Preparation

pCAMBIA2300::35S::DsRed2 (Appendix A), composed of a reporter gene, *DsRed2*, driven by a 35S promoter, which was gifted to us from Prof. ShuangXia Jin of Huazhong Agricultural University (Wuhan, China), was transferred into *A. tumefaciens* LBA4404. Positive clones were streaked onto LB solid medium supplemented with 20 μg/mL rifampicin and 50 μg/mL kanamycin and then incubated at 28 °C for two days. The bacterial lawn was scraped using an inoculation ring and inoculated into MGL liquid medium (5.0 g/L peptone, 0.25 g/L KH_2_PO_4_, 5 g/L NaCl, 0.1 g/L MgSO_4_ 7H_2_O, 1.0 g/L glycine, 5.0 g/L D-mannitol) containing 30 mg/mL acetosyringone. The bacterial solution was cultivated using a rotatory shaker at 200 rpm and 28 °C for four hours until the OD_600_ reached 0.8, which was used for transformation.

### 4.3. Plant Preparation and Infection

For perennial plants like *L. regale* and *P. notoginseng*, the aboveground parts will wither during winter. When the next spring arrives, the tender shoot (stem and leaves) will sprout again from the meristems of its rhizome or bulb and grow into a complete plant. At the same time, some new tuberous roots are generated from the *P. notoginseng* old taproot, and some new roots and scales also generate from the meristems in the *L. regale* bulb.

In spring, the soil above the *P. notoginseng* rhizome was gently removed before the germination of tender shoots. *A. tumefaciens* LBA4404 containing pCAMBIA2300::35S::DsRed2 was injected into the rhizome of *P. notoginseng* using a syringe at two positions with one injection per position (Appendix A). A total of 5 μL of bacterial solution was injected into each position (Figure 1). Then, the soil was moved so that it gently covered the rhizome. In total, 71 *P. notoginseng* rhizomes were injected with *A. tumefaciens*.

Similarly, one-year-old *L. regale* bulbs were collected from the soil before sprouting. A 5 μL bacterial solution was sucked with a syringe, and the needle was vertically inserted into the meristems of the *L. regale* bulb to inject the *A. tumefaciens* (Figure 3 and Appendix A). After this injection, each *L. regale* bulb was cultured at 28 °C in the dark for 2 days before being planted in the nutrient soil. In total, 82 *L. regale* bulbs were subjected to injection.

### 4.4. PCR Screening of Positive Transgenic P. notoginseng and L. regale Plants

New seedlings of *P. notoginseng* and *L. regale* (candidate transgenic plant) were produced after injection. One leaf of one candidate plant was collected for genomic DNA extraction. *DsRed2*-specific primers (F: 5′-ACAGAACTCGCCGTAAAGAC-3′; R: 5′-CCGTCCTCGAAGTTCATCAC-3′) were used for PCR, with the genomic DNA serving as the template. The PCR reaction mix consisted of 12.5 μL of 2× GS Taq PCR Mix, 0.2 μg genomic DNA, 0.1 μL forward primer (10 μM), 0.1 μL reverse primer (10 μM), and double-distilled water (total volume: 25 μL). The PCR conditions were as follows: 94 °C for 5 min; 94 °C for 30 s, 56 °C for 30 s, and 72 °C for 12 s for 32 cycles; and 72 °C for 10 min. The PCR products were analyzed using 1.2% agarose gel electrophoresis. Moreover, the PCR products from the genomic DNA of the wild-type *P. notoginseng* and *L. regale* leaves, as well as the pCAMBIA2300::35S::DsRed2 plasmids, were used as negative and positive controls, respectively.

### 4.5. qRT-PCR-Based Detection of DsRed2 Expression

Total RNA was extracted from the leaves and stems of positive transgenic *P. notoginseng* plants, as well as the leaves of transgenic *L. regale*, and then the RNA was reverse-transcribed into cDNA. qRT-PCR was performed as described by Su et al. [29] using *DsRed2*-specific primers (F: 5′-CCAGTTCCAGTACGGCTCCAA-3′; R: 5′-AGGAGTCCTGGGTCACGGTC-3′), and the relative expression level of *DsRed2* was calculated using the −2^ΔΔCt^ method [30].

### 4.6. Analysis of Red Fluorescent Protein Expression through Western Blot

Total protein was extracted from the leaves of transgenic *P. notoginseng* and *L. regale* using a plant total protein extraction kit (Sangon, Shanghai, China). The concentration of total protein was quantified at 10 μg/μL and subjected to sodium dodecyl sulfate polyacrylamide gel electrophoresis (SDS-PAGE) using a 5% concentrated gel and a 12% separating gel. Electrophoresis was performed at 80 V for 40 min, followed by 120 V for 80 min.

A polyvinylidene fluoride (PVDF) membrane (0.45 μm, Sigma-Aldrich, St. Louis, MO, USA) was used to transfer the proteins from the gel using a current of 400 mA in an ice bath for 60 min. After membrane transfer, the PVDF membrane was divided into upper and lower parts. The upper membrane, containing proteins with a molecular weight greater than 45 kDa, was incubated with the primary antibody (Affinity, Shanghai, China) against the actin protein, serving as an internal reference protein. The lower half of the membrane, containing proteins with a molecular weight less than 45 kDa, was incubated with the primary antibody (Affinity, China) against the red fluorescent protein encoded by *DsRed2*. Both membranes were incubated in a rotary shaker at 80 rpm and 28 °C for 2 h, followed by two washes with 1X TBST (Tris-HCl with Tween-20) for 10 min each. Subsequently, the membranes were incubated with the secondary antibody (anti-mouse IgG H&L/HRP antibody, Affinity, China) in a rotary shaker at 80 rpm and 28 °C for 1 h. After incubation, the membranes were washed twice with 1X TBST for 10 min each. Finally, superKine™ ultra-sensitive ECL luminescent liquid (Abbkine Scientific, Wuhan, China) was used for visualization, and exposure was performed using a transilluminator (Bio-Rad, Hercules, CA, USA).

### 4.7. Observation of Specific Red Fluorescence in Different Organs of DsRed2 Transgenic Plants Using Laser Confocal Microscopy

Three transgenic plants with a high expression of *DsRed2* were selected for fluorescence observation. To confirm the biological activity of the red fluorescent protein expressed in the transgenic plants, the leaves, stems, and newly grown tuberous roots of *DsRed2* transgenic *P. notoginseng* plants were collected. Subsequently, they were sliced using a surgical knife. The samples were placed on a glass slide and covered with a cover slip before being observed using a laser confocal microscope (Nikon, Tokyo, Japan) at an excitation wavelength of 560 nm and emission wavelength of 575 nm [17]. Similarly, the leaves, stems, newly grown roots and scales of *DsRed2* transgenic *L. regale* plants were harvested for fluorescence observation using the same method.

### 4.8. Observing Red Fluorescence in Transgenic Plants Using a Handheld Fluorescence Detector

In the second year after injection, wild-type plants, as well as the transgenic plants without additional injection of *A. tumefaciens* LBA4404 that grew normally and had a high expression of *DsRed2*, were collected to observe the bright orange fluorescence derived from DsRed2 expression using handheld fluorescence observation equipment (LUYOR-3415RG, Shanghai, China), with a green light indicating excitation [18]. Fluorescence observations were recorded with a camera (Canon, Tokyo, Japan) with a red filter lens.

## 5. Conclusions

The tissue-culture-independent genetic transformation method presented in this study is a fast and efficient transformation method for *P. notoginseng* and *L. regale*. The reporter gene *DsRed2* was successfully integrated into the genome of *P. notoginseng* and *L. regale* and was normally expressed in multiple organs. The red fluorescence of DsRed2 was detected in transgenic *P. notoginseng* and *L. regale*. Most importantly, the *DsRed2* transgenic *P. notoginseng* and *L. regale* plants still maintained red fluorescence activity in the second year after injection. This transformation method is suitable for plants with a low regeneration ability or those with a low transformation rate, such as *P. notoginseng* and *L. regale.*

## Figures and Tables

**Figure 1 plants-13-02509-f001:**
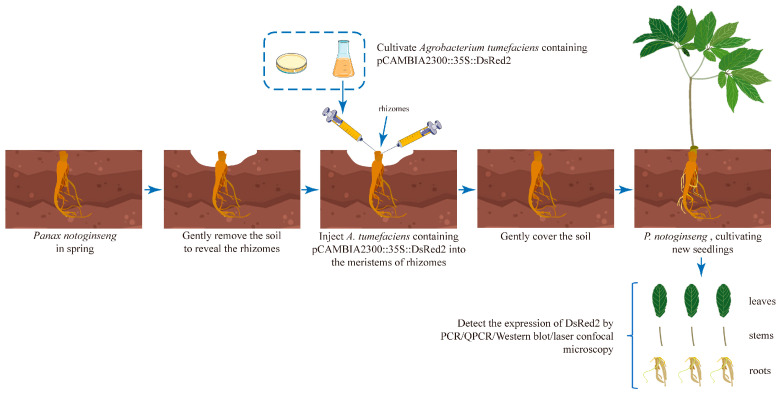
Schematic diagram illustrating *Agrobacterium tumefaciens* injection into the rhizome of *Panax notoginseng*.

**Figure 2 plants-13-02509-f002:**
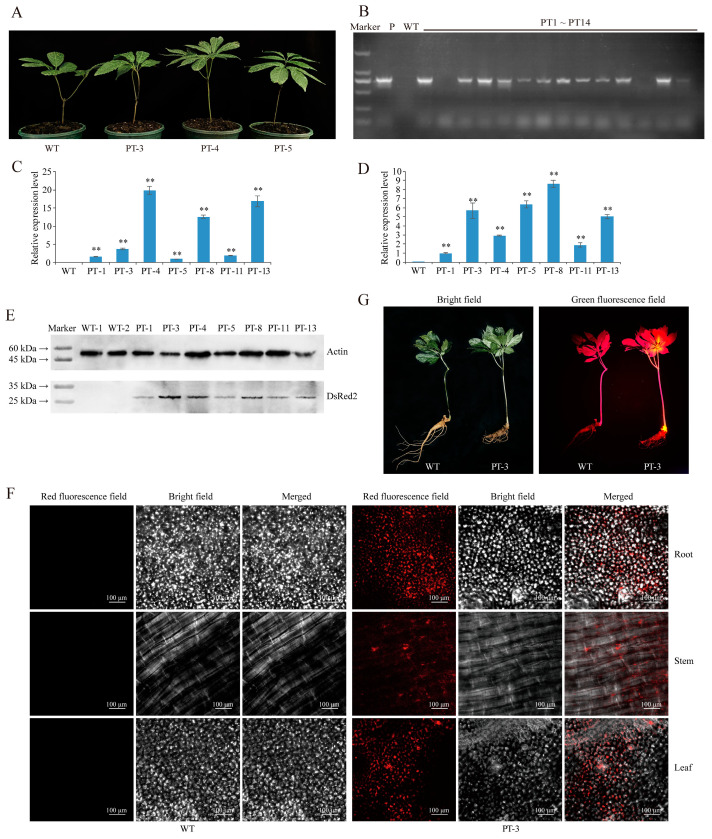
*A. tumefaciens* injection into the rhizome of *P. notoginseng* is a stable and efficient transformation method. (**A**) Comparison of phenotype between the injected *P. notoginseng* plants (PT-3/4/5) and wild-type (WT) plant. (**B**) Gel electrophoresis showing *DsRed2* (*Discosoma striata red fluorescence protein 2*) amplification from genomic DNA of injected *P. notoginseng* leaves. P, WT, and PT1-PT14 represent PCR reaction of wild-type *P. notoginseng*, pCAMBIA2300S-*DsRed2* plasmids, and *DsRed2* transgenic *P. notoginseng*, respectively. (**C**) Expression of *DsRed2* in leaves of transgenic *P. notoginseng* as determined by qPCR. WT and PT-1/3/4/5/8/11/13 represent the qPCR reaction with the leaves of wild-type and *DsRed2* transgenic *P. notoginseng* plants, respectively. (**D**) Expression of *DsRed2* in the stems of transgenic *P. notoginseng* as determined by qPCR. WT and PT-1/3/4/5/8/11/13 represent the qPCR reaction with the stems of wild-type and *DsRed2* transgenic *P. notoginseng* plants, respectively. (**E**) Detection of DsRed2 protein expression in leaves of transgenic *P. notoginseng* through Western blot. WT-1/2 and PT-1/3/4/5/8/11/13 represent the leaf total protein of wild-type and *DsRed2* transgenic *P. notoginseng* plants, respectively. (**F**) Observation of specific red fluorescence in leaves, stems, and roots of transgenic *P. notoginseng* under a laser scanning confocal microscope. WT and PT-3 represent the wild-type and *DsRed2* transgenic *P. notoginseng* plants, respectively. (**G**) Observation of specific fluorescence in transgenic *P. notoginseng* with a handle fluorescence detector. WT and PT-3 represent the wild-type and *DsRed2* transgenic *P. notoginseng* plants in the second year after injection, respectively. *T*-test was used to reveal the statistical difference of *DsRed2* expression level between wild type and transgenic plants (** *p* < 0.01).

**Figure 3 plants-13-02509-f003:**
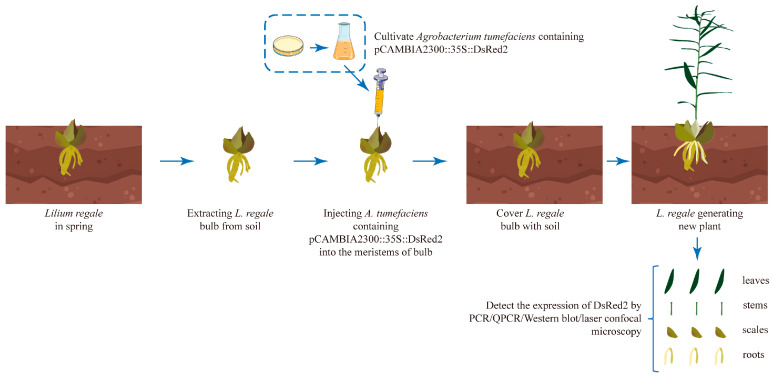
Schematic diagram illustrating *A. tumefaciens* injection into the meristems of *Lilium regale* bulb.

**Figure 4 plants-13-02509-f004:**
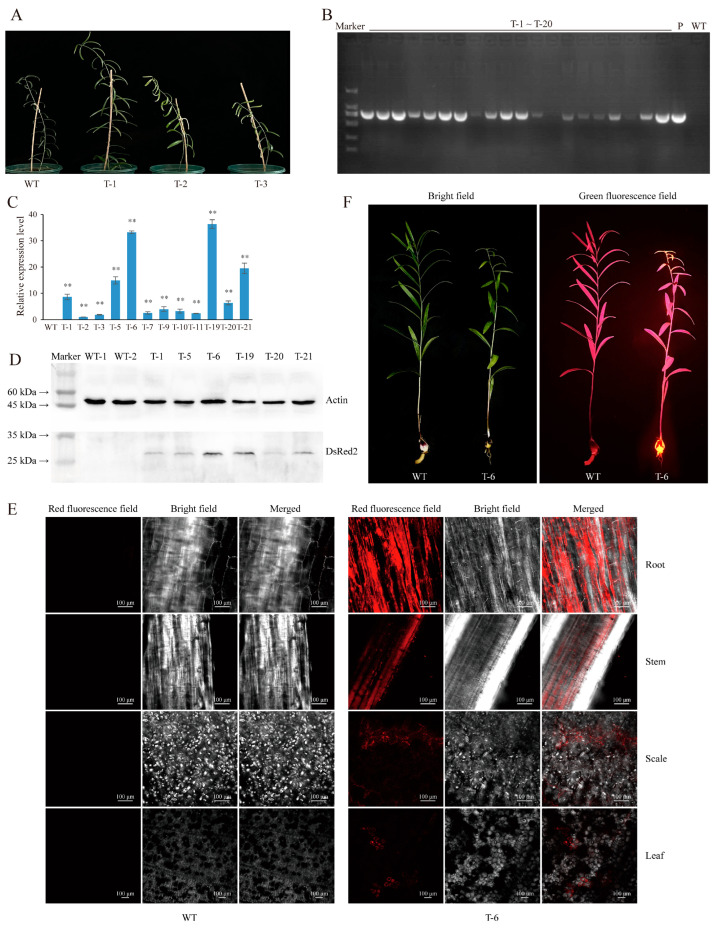
*A. tumefaciens* injection into the meristems of the *L. regale* bulb is a fast and efficient transformation method. (**A**) Comparison of phenotype between the injected *L. regale* plants (T-1/2/3) and wild-type (WT) plant. (**B**) Gel electrophoresis showing *DsRed2* amplification from injected *L. regale* leaves. T-1~T-20, P, and WT represent PCR reaction of *DsRed2* transgenic *L. regale*, pCAMBIA2300S-*DsRed2* plasmids, and wild-type, respectively. (**C**) Expression of *DsRed2* in leaves of transgenic *L. regale* as determined by qPCR. WT and T-1/2/3/5/6/7/9/10/11/19/20/21 represent qPCR reaction of the wild-type and *DsRed2* transgenic *L. regale* plants. (**D**) Detection of DsRed2 protein expression in leaves of transgenic *L. regale* through Western blot. WT and T-1/5/6/19/20/21 represent the leaf total protein of wild-type and *DsRed2* transgenic *L. regale* plants. (**E**) Observation of specific red fluorescence in leaves, stems, roots, and scales of transgenic *L. regale* under a laser scanning confocal microscope. WT and T-6 represent the wild-type and *DsRed2* transgenic *L. regale* plants, respectively. (**F**) Observation of specific fluorescence in transgenic *L. regale* with a handle fluorescence detector. WT and T-6 represent the wild-type and *DsRed2* transgenic *L. regale* in the second year after injection, respectively. *T*-test was used to reveal the statistical difference of *DsRed2* expression level between wild type and transgenic plants (** *p* < 0.01).

## Data Availability

Data are contained within the article and Appendix A.

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
