# Peer review of "A Fast, Efficient, and Tissue-Culture-Independent Genetic Transformation Method for Panax notoginseng and Lilium regale"

_plants, 2024, doi:10.3390/plants13172509_

Round 1

Reviewer 1 Report

Comments and Suggestions for Authors

Comments and suggestions for authors:

1. Introduction section

lines 41 to 50: In this paragraph the authors indicate the difficulties of in vitro culture and genetic transformation in both botanical genera, but they should include those more recent works where it has been possible to develop a genetic transformation protocol, both in Panex and in Lilium, for example, in Panex the works of Choi et al 2022, Liu et al 2024, Jiang et al, 2024, and in Lilium that of Fu et al 2023.

lines 55-59: Include in this paragraph the work of Zhang et al 2019 (J. Agric. Food Chem. 2019, 67, 1982−1989).

lines 65-74: describe in this paragraph the work on lilium somatic embryogenesis ((Sun et al 2023), the work on transient pollen transformation (Zhang et al, 2023) and the regeneration and transformation of Lilium brownii (Fu et al. al, 2023, Plants (Basel), 12(10):1992. doi: 10.3390/plants12101992. PMID: 37653909) so that the introduction is more complete and better reflects the difficulties of genetic transformation of Lilium spp.

Line 75-include a paragraph talking about the DSRed2 reporter gene and include references about this marker.

2. Material and methods

This section is not well structured and the experiments are missing important information. I suggest you follow a scheme as indicated or similar:

4.1. Plant material - include information about the plant material here: what origin do the plants have, were they obtained from seed germination, were they purchased? What growing conditions did they have, substrate, temperature and lighting in the greenhouse.

4.2. A. tumefaciens strain, binary vector and culture preparation- Indicating the origin of the Agrobacterium strain and describing the vector used, it would help a vector diagram indicating the genes within the T-DNA area where you can see which promoter the DsRed2 gene has. Did you check different Agrobacterium strains? Indicate what MGL medium is.

4.3. Plant preparation and infection- This section should describe in more detail both how the plant material is prepared and how the infection is carried out. For example, there are information missing, such as: what growing conditions both the rhizome and the bulb had? How many rhizomes and bulbs were used? How many times was the experiment repeated? Additionally, it should be indicated how much Agrobacterium solution was injected? At a single point or at several points? Was only one application made or several? And better explain what the growth point of the rhizome and bulb is. On the other hand, the way in which the infection is carried out can raise doubts about whether solid transformants or chimera plants are obtained and about its effectiveness. In other papers, a cut is made and the entire cut area is infected, thus favouring the regenerated shoots to be transgenic with greater probability, but in this work, it is an injection at one point. Therefore, we could ask if there were not many escapes and chimeras. Clarify this point in more detail.

4.4 (4.2, lines 238-248). Include the sequence of the primers used in the text and not in the supplementary material. PCR was done on separated leaves from each putative plant or on a mixture of leaves?

4.5 (4.3, lines 250-254). Add that the primer sequences are indicated in the previous section (in line 253) and add to the end of the section the reference of the method for calculating the expression level of the genes.

Line 278-289: Explain better how the plants were observed under the microscope. Was a red filter used? At what wavelength of excitation and emission? Indicate that only one transgenic plant and an untransformed control were used for this study. 

3. Result section.

Line 94-97: This paragraph is not clear, were 71 rhizomes injected and 71 plants obtained? Were plants obtained from all the rhizomes? Only one plant is produced per rhizome?

Line 104-108: only one plant of the 61 transgenic plants obtained was studied by microscopy?

In figure 2F, you can see how the control plants under red fluorescence field appear all dark and nothing is seen, however, in figure 2G the control plant under red fluorescence field also looks red when it should be seen dark. How is this explained? As a handle fluorescence detector was used, the light it emits is possibly green light. Explain all this in greater detail.  

Lines 145-146: indicate if 82 bulbs were injected, and if 82 plants were obtained from them. You only get one plant per injected bulb? Line 145 is not well written: "showed specific products"??

Figure 4. Same as in figure 2. Fifure 4F: Check is the field was green light (see Huai et al (2023) Red fluorescence protein (DsRed2) promotes the screening efficiency in peanut genetic transformation. Front. Plant Sci. 14:1123644. doi: 10.3389/fpls.2023.1123644).

-Include the time it takes to obtain transgenic plants of both species with the developed method.

4. Discussion section.

-The authors could also discuss the time it takes to obtain a transgenic plant with standard methods and compare it with the times obtained using the developed method.

Author Response

Dear reviewer,

We have completely revised our manuscript "a fast, efficient, and tissue-culture independent genetic trans-formation method of Panax notoginseng and Lilium regale". MDPI English Editing further revised the English editing of this manuscript. The responses to the comments are as follows:

  1. Introduction section

Comments 1: lines 41 to 50: In this paragraph the authors indicate the difficulties of in vitro culture and genetic transformation in both botanical genera, but they should include those more recent works where it has been possible to develop a genetic transformation protocol, both in Panex and in Lilium, for example, in Panax the works of Choi et al 2022, Liu et al 2024, Jiang et al, 2024, and in Lilium that of Fu et al 2023.

Response 1: Thanks for your advice, and the recent genetic transformation researches of Panax spp. and Lilium spp. were added in the introduction.

Panax spp., including P. notoginseng, are well known medicinal herbs with valuable medicinal properties, primarily attributed to their saponin content [5]. However, the lack of a well-established genetic transformation system has hindered further exploration of the saponin biosynthesis pathways and the development of elite P. notoginseng cultivars with high and consistent yields and quality [6]. Adventitious roots were regenerated from Panax ginseng roots through tissue culture and transgenic P. ginseng roots were obtained via Agrobacterium-based transformation in [7]. In another study, the RNA interference of PgRg1-3 in the transgenic P. ginseng roots led to genetic transformation mediated by Agrobacterium rhizogenes C58C1 [8]. The PgERF120 was overexpressed in the hairy roots of P. ginseng roots based on Agrobacterium tumefaciens A4 transformation in [9]. Elsewhere, P. notoginseng cell lines overexpressing Panax japonicus βAS (β‐amyrin synthase) were obtained via Agrobacterium-mediated genetic transformation in order to reveal the influence of PjβAS on saponin biosynthesis in P. notoginseng [10]. The current transformation method involves traditional Agrobacterium-based techniques applied to callus tissue induced from P. notoginseng explants, but successful generation of transgenic P. notoginseng plants has not been achieved using this approach [11]. At the same time, aseptic operation makes the acquisition of genetically modified materials more time-consuming and more difficult.

Similarly, Lilium spp., known for their wide range of flower shapes, fragrances, and colors, hold significant ornamental value and rank as the sixth-largest fresh cut flower globally [12]. However, the traditional Agrobacterium-mediated transformation method, characterized by its low efficiency, lengthy cycle, and demanding sterile operation, is not widely applicable in molecular breeding efforts for lilies [13]. In another study, although the transformation rate of Lilium ‘Sorbonne’ was improved by adjusting the composition of the culture medium, only 14 positive transgenic Lilium ‘Sorbonne’ lines out of 173 lines were successfully identified [14]. Elsewhere, the transformation rate of Lilium brownie embryogenic callus was found to be 65.56% [15]. Also in another study, the β-glucuronidase (GUS) gene and green fluorescent protein(GFP) gene were successfully expressed in L. regale via pollen magnetofection, but a complete transgenic plant was not produced [16].Therefore, the transformation rate of lilies is still low, and the protocols remain difficult and complex. L. regale, a wild lily species with high resistance to various stresses, remains underexplored due to the lack of a mature transformation system, hindering the elucidation of its resistance mechanisms and the utilization of its resistance-related genes to improve lily disease resistance. Establishing an efficient genetic transformation system is crucial for enhancing the ornamental and disease-resistant traits of lilies to improve the economic value of Lilium spp.

Comments 2: lines 55-59: Include in this paragraph the work of Zhang et al 2019 (J. Agric. Food Chem. 2019, 67, 1982−1989).

Response 2: Thanks for your advice, and the reference was added in the introduction.

Panax spp., including P. notoginseng, are well known medicinal herbs with valuable medicinal properties, primarily attributed to their saponin content [5]. However, the lack of a well-established genetic transformation system has hindered further exploration of the saponin biosynthesis pathways and the development of elite P. notoginseng cultivars with high and consistent yields and quality [6]. Adventitious roots were regenerated from Panax ginseng roots through tissue culture and transgenic P. ginseng roots were obtained via Agrobacterium-based transformation in [7]. In another study, the RNA interference of PgRg1-3 in the transgenic P. ginseng roots led to genetic transformation mediated by Agrobacterium rhizogenes C58C1 [8]. The PgERF120 was overexpressed in the hairy roots of P. ginseng roots based on Agrobacterium tumefaciens A4 transformation in [9]. Elsewhere, P. notoginseng cell lines overexpressing Panax japonicus βAS (β‐amyrin synthase) were obtained via Agrobacterium-mediated genetic transformation in order to reveal the influence of PjβAS on saponin biosynthesis in P. notoginseng [10]. The current transformation method involves traditional Agrobacterium-based techniques applied to callus tissue induced from P. notoginseng explants, but successful generation of transgenic P. notoginseng plants has not been achieved using this approach [11]. At the same time, aseptic operation makes the acquisition of genetically modified materials more time-consuming and more difficult.

Comments 3: lines 65-74: describe in this paragraph the work on lilium somatic embryogenesis ((Sun et al 2023), the work on transient pollen transformation (Zhang et al, 2023) and the regeneration and transformation of Lilium brownii (Fu et al. al, 2023, Plants (Basel), 12(10):1992. doi: 10.3390/plants12101992. PMID: 37653909) so that the introduction is more complete and better reflects the difficulties of genetic transformation of Lilium spp.

Response 3: Thanks for your advice, and these articles were added in the introduction.

Similarly, Lilium spp., known for their wide range of flower shapes, fragrances, and colors, hold significant ornamental value and rank as the sixth-largest fresh cut flower globally [12]. However, the traditional Agrobacterium-mediated transformation method, characterized by its low efficiency, lengthy cycle, and demanding sterile operation, is not widely applicable in molecular breeding efforts for lilies [13]. In another study, although the transformation rate of Lilium ‘Sorbonne’ was improved by adjusting the composition of the culture medium, only 14 positive transgenic Lilium ‘Sorbonne’ lines out of 173 lines were successfully identified [14]. Elsewhere, the transformation rate of Lilium brownie embryogenic callus was found to be 65.56% [15]. Also in another study, the β-glucuronidase (GUS) gene and green fluorescent protein(GFP) gene were successfully expressed in L. regale via pollen magnetofection, but a complete transgenic plant was not produced [16].Therefore, the transformation rate of lilies is still low, and the protocols remain difficult and complex. L. regale, a wild lily species with high resistance to various stresses, remains underexplored due to the lack of a mature transformation system, hindering the elucidation of its resistance mechanisms and the utilization of its resistance-related genes to improve lily disease resistance. Establishing an efficient genetic transformation system is crucial for enhancing the ornamental and disease-resistant traits of lilies to improve the economic value of Lilium spp.

Comments 4: Line 75-include a paragraph talking about the DsRed2 reporter gene and include references about this marker.

Response 4: Thanks for your advice, and the introduction of DsRed2 was added in the revised manuscript.

DsRed2 is a red fluorescent protein isolated from Discosoma sp. that, in another study, was found to be a better reporter gene than GUS and GFP because its excitation light was higher than that of GUS and GFP; thus, it can avoid plant autofluorescence [17]. DsRed2 has been used to screen transgenic cotton (Gossypium herbaceum) and transgenic peanut (Arachis hypogaea) via fluorescence microscopy and handheld fluorescence observation equipment through the detection of red and bright orange fluorescence, respectively [17, 18]. To address the challenges associated with applying the transitional Agrobacterium-based transformation method to perennial plants such as Panax spp. and Lilium spp., we utilized P. notoginseng and L. regale as two experimental models to establish a rapid, efficient, and tissue-culture-independent genetic transformation system. Agrobacterium tumefaciens of the LBA4404 strain containing the vector pCAMBIA2300::35S::DsRed2 were injected into the meristems of P. notoginseng rhizome and L. regale bulb. Through 40-60 days of cultivation after Agrobacterium injection, a series of experiments were conducted to evaluate the expression levels and activity of the reporter gene in transgenic plants.

  1. Material and methods

This section is not well structured and the experiments are missing important information. I suggest you follow a scheme as indicated or similar:

Comments 5: 4.1. Plant material - include information about the plant material here: what origin do the plants have, were they obtained from seed germination, were they purchased? What growing conditions did they have, substrate, temperature and lighting in the greenhouse.

Response 5: Thanks for your advice. We have added a section (4.1. Plant material) in our manuscript. This section introduced the generating methods of the P. notogiseng and L. regale, and it also contains the details of growth conditions.

4.1 Plant materials

After collecting from three-year-old plants, the coat of P. notoginseng seeds were removed and sown in damp sands with 20%-30% humidity before being placed in a refrigerator (8°C) for low-temperature treatment. After 30 days, the seeds were sown in a shed with a sunshade net. In addition, the L. regale seeds were sown in nutrient soil (vermiculite: perlite: nutrient soil=2:1:1) and cultivated in a greenhouse (25°C, nature light). One-year-old P. notoginseng and L. regale plants were used for gene transformation.

Comments 6: 4.2. A. tumefaciens strain, binary vector and culture preparation- Indicating the origin of the Agrobacterium strain and describing the vector used, it would help a vector diagram indicating the genes within the T-DNA area where you can see which promoter the DsRed2 gene has. Did you check different Agrobacterium strains? Indicate what MGL medium is.

Response 6: Thanks for your advice. We have added a section (4.2. A. tumefaciens strain, binary vector and culture preparation) in our manuscript. The schematic representation of the DsRed2 vector was added in supplementary materials as Figure S3. The DsRed2 was driven by promoter CaMV35S. We only used A. tumefaciens LBA4404 in this study, and the other Agrobacterium strains were not checked. At the same time, the formula for MGL medium was also added in this section.

4.2 A. tumefaciens strain, binary vector, and culture preparation

pCAMBIA2300::35S::DsRed2 (Figure S3), composed of a reporter gene, DsRed2, driven by a 35S promoter, which was gifted to us from Pro. ShuangXia Jin of Huazhong Agricultural University, was transferred into A. tumefaciens LBA4404. Positive clones were streaked onto LB solid medium supplemented with 20 μg/mL rifampicin and 50 μg/mL kanamycin and then incubated at 28°C for two days. The bacterial lawn was scraped using an inoculation ring and inoculated into MGL liquid medium (5.0 g/L peptone, 0.25 g/L KH2PO4, 5 g/L NaCl, 0.1 g/L MgSO4 7H2O, 1.0 g/L glycine, 5.0 g/L D-mannitol) containing 30 mg/mL acetosyringone. The bacterial solution was cultivated using a rotatory shaker at 200 rpm and 28°C for four hours until the OD600 reached 0.8, which was used for transformation.

Comments 7: 4.3. Plant preparation and infection- This section should describe in more detail both how the plant material is prepared and how the infection is carried out. For example, there are information missing, such as: what growing conditions both the rhizome and the bulb had? How many rhizomes and bulbs were used? How many times was the experiment repeated? Additionally, it should be indicated how much Agrobacterium solution was injected? At a single point or at several points? Was only one application made or several? And better explain what the growth point of the rhizome and bulb is. On the other hand, the way in which the infection is carried out can raise doubts about whether solid transformants or chimera plants are obtained and about its effectiveness. In other papers, a cut is made and the entire cut area is infected, thus favoring the regenerated shoots to be transgenic with greater probability, but in this work, it is an injection at one point. Therefore, we could ask if there were not many escapes and chimeras. Clarify this point in more detail.

Response 7: Thanks for your advice. We have added a section (4.3. Plant preparation and infection) in the manuscript. The contents you recommended added in the manuscript were explained in section 4.3 to further improve the introduction of ‘Methods’. In addition, the discussion about the transgenic plants generated by this transformation method whether is a chimera plant was added.

4.3 Plant preparation and infection

For perennial plants like L. regale and P. notoginseng, the aboveground parts will wither during winter. When the next spring arrives, the tender shoot will sprout again from its meristem of rhizome or bulb and grow into a complete plant. In spring, the soil above the P. notoginseng rhizome was gently removed before the germination of tender shoots. A. tumefaciens LBA4404 containing pCAMBIA2300::35S::DsRed2 was injected into the rhizome of P. notoginseng twice using a syringe at two positions (Figure S4). A total of 5 μL of bacterial solution was injected into each position (Figure 1). Then, the soil was moved so that it gently covered the rhizome. In total, 71 P. notoginseng rhizomes were injected with A. tumefaciens.

Similarly, one-year-old L. regale bulbs were collected from the soil before sprouting. A 5 μL bacterial solution was sucked through a syringe, and the needle was vertically inserted into the meristems of L. regale bulb to inject the A. tumefaciens (Figure 3 and Figure S5). After this injection, each L. regale bulb was cultured at 28°C in the dark for 2 days before being planted in the nutrient soil. In total, 82 L. regale bulbs were subjected to injection.

  1. Discussion

Recently, a novel Agrobacterium-based transformation method that forgoes the need for sterile operations, developed by Cao et al. [23], has been widely adopted for transgenic plants with high redifferentiation capabilities, such as herbaceous and tuberous plants. However, transformation methods for plants with low differentiation abilities require further exploration. In this study, we introduced a fast and efficient Agrobacterium transformation method based on the injection of A. tumefaciens LBA4404 containing DsRed2 into the meristems of L. regale bulb and P. notoginseng rhizome. This method eliminates the need for sterile operations and meets the requirements for plants with low regeneration abilities. The successful expression of the reporter gene DsRed2 was demonstrated in multiple organs of the P. notoginseng and L. regale plants (Figure 2 and Figure 4). Protein expression of DsRed2 was confirmed via Western blot analysis, and the presence of red fluorescence in multiple organs of positive transgenic P. notoginseng and L. regale plants indicated normal activity of the red fluorescent protein (Figure 2E, F and Figure 4D, E). Most importantly, in the second year after injection, specifically, bright orange fluorescence was observed in the transgenic P. notoginseng and L. regale plants (Figure 2G and Figure 4F), evidencing the stable expression of DsRed2. This study provides robust technical support for fundamental research on plants with low transformation rates and low regeneration abilities. In addition, as perennial plants, it takes 2-3 and 6-8 years for P. notoginseng and L. regale to generate seeds, respectively; therefore, the transgenic plants featured in this study are the chimera plants. Nonetheless, the genetic features of DsRed2 require further investigation in order to clearly understand the stability of the transgenic method established in this study.

Comments 8: 4.4 (4.2, lines 238-248). Include the sequence of the primers used in the text and not in the supplementary material. PCR was done on separated leaves from each putative plant or on a mixture of leaves?

Response 8: Thanks for your advice. The sequences of primers for PCR were added in the ‘Methods’. The PCR template was the genome DNA of the leaves of each putative plant. The content we revised or added was highlighted in red.

4.4 PCR Screening of positive transgenic P. notoginseng and L. regale plants

New seedlings of P. notoginseng and L. regale (candidate transgenic plant) were produced after injection. One leaf of one candidate plant was collected for genomic DNA extraction. DsRed2-specific primers (F: 5’-ACAGAACTCGCCGTAAAGAC-3’; R: 5’-CCGTCCTCGAAGTTCATCAC-3’) were used for PCR, with the genomic DNA serving as the template. The PCR reaction mix consisted of 12.5 μL of 2× GS Taq PCR Mix, 0.2 μg genomic DNA, 0.1 μL forward primer (10 μM), 0.1 μL reverse primer (10 μM), and double-distilled water (total volume: 25 μL). The PCR conditions were as follows: 94°C for 5 min; 94°C for 30 s, 56°C for 30 s, and 72°C for 12 s for 32 cycles; and 72°C for 10 min. The PCR products were analyzed using 1.2% agarose gel electrophoresis. Moreover, the PCR products from the genomic DNA of the wild-type P. notoginseng and L. regale leaves, as well as the pCAMBIA2300::35S::DsRed2 plasmids, were used as negative and positive controls, respectively.

Comments 9: 4.5 (4.3, lines 250-254). Add that the primer sequences are indicated in the previous section (in line 253) and add to the end of the section the reference of the method for calculating the expression level of the genes.

Response 9: Thanks for your advice. The sequences of primers for qRT-PCR was added in the ‘Methods’. A reference about the calculating methods was also added.

4.5 qRT-PCR-based Detection of DsRed2 expression

Total RNA was extracted from the leaves and stems of positive transgenic P. notoginseng plants, as well as the leaves of transgenic L. regale, and then the RNA was reverse-transcribed into cDNA. qRT-PCR was performed as described by Su et al. [27] using DsRed2-specific primers (F: 5’-CCAGTTCCAGTACGGCTCCAA-3’; R: 5’-AGGAGTCCTGGGTCACGGTC-3’), and the relative expression level of DsRed2was calculated using the -2ΔΔCt method [28].

Comments 10: Line 278-289: Explain better how the plants were observed under the microscope. Was a red filter used? At what wavelength of excitation and emission? Indicate that only one transgenic plant and an untransformed control were used for this study.

Response 10: Thanks for your advice. The details of microscope observation methods were added in the ‘Methods’. For laser confocal microscopy (Nikon, Japan), the organs of the transgenic plant were sliced and observed in a specific excitation wavelength (560 nm). For handheld fluorescence observation equipment (LUYOR-3415RG, USA), a completed transgenic plant was observed with green light, and the camera used for photos was equipped with a red filter. In addition, the qRT-PCR and western blot were used to screen the transgenic plants that has high DsRed2 expressed. The transgenic plants after screened by qRT-PCR and western blot that have high DsRed2 expression were used for microscope observation. Finally, two transgenic plants that had excellent fluorescence in the second year after injection were included in the manuscript. Moreover, the fluorescence of another transgenic P. notogisneng plant PT-4 was observed, the result was shown in Figure 1 of response list.

Figure 1 In the second year after injection, the fluorescence observation of PT-4 by a handheld fluorescence observation equipment.

4.7. Observation of specific red fluorescence in different organs of DsRed2 transgenic plants using laser confocal microscopy

Three transgenic plants with a high expression of DsReds were selected for fluorescence observation. To confirm the biological activity of the red fluorescent protein expressed in the transgenic plants, the leaves, stems, and roots of DsRed2 transgenic P. notoginseng plants were collected. Subsequently, they were sliced using a surgical knife. The samples were placed on a glass slide and covered with a cover slip before being observed using a laser confocal microscope (Nikon, Japan) at an excitation wavelength of 560 nm and emission wavelength of 575 nm [17]. Similarly, the leaves, stems, roots, and scales of DsRed2 transgenic L. regale plants were harvested for fluorescence observation using the same method.

4.8. Observing red fluorescence in transgenic plants using a handheld fluorescence detector

In the second year after injection, wild-type plants, as well as transgenic plants that grew normally and had a high expression of DsRed2, were collected to observe the bright orange fluorescence derived from DsRed2 expression using handheld fluorescence observation equipment (LUYOR-3415RG, USA), with a green light indicating excitation [18]. Fluorescence observations were recorded with a camera (Canon, Japan) with a red filter lens.

  1. Result section

Comments 11: Line 94-97: This paragraph is not clear, were 71 rhizomes injected and 71 plants obtained? Were plants obtained from all the rhizomes? Only one plant is produced per rhizome?

Response 11: Thanks for your advice. P. notogiseng was perennial plant, and one P. notogiseng has one rhizome. In spring, a shoot sprouts from the rhizome and develops the aboveground part (stem, leaf, flower, and seed) of P. notogiseng plant. A total 71 rhizomes were injected by A. tumefaciens LBA4404 containing DsRed2, and 71 rhizomes generated new shoots. These contents were also added in the manuscript and highlighted in red.

4.3 Plant preparation and infection

For perennial plants like L. regale and P. notoginseng, the aboveground parts will wither during winter. When the next spring arrives, the tender shoot will sprout again from its meristem of rhizome or bulb and grow into a complete plant. In spring, the soil above the P. notoginseng rhizome was gently removed before the germination of tender shoots. A. tumefaciens LBA4404 containing pCAMBIA2300::35S::DsRed2 was injected into the rhizome of P. notoginseng twice using a syringe at two positions (Figure S4). A total of 5 μL of bacterial solution was injected into each position (Figure 1). Then, the soil was moved so that it gently covered the rhizome. In total, 71 P. notoginseng rhizomes were injected with A. tumefaciens.

Comments 12: Line 104-108: only one plant of the 61 transgenic plants obtained was studied by microscopy?

Response 12: Thanks for your advice. A total of 61 transgenic plants were screened by PCR, but two transgenic plants that has high DsRed2 expression (after qRT-PCR and western blot analysis) in the second year after injection were used for fluorescence observation. A P. notoginseng transgenic plant (PT-3) that had excellent fluorescence in the second year after injection were included in the manuscript. Moreover, the fluorescence of another transgenic P. notogisneng plant PT-4 was observed, the result was shown in Figure 1 of response list.

Comments 13: In figure 2F, you can see how the control plants under red fluorescence field appear all dark and nothing is seen, however, in figure 2G the control plant under red fluorescence field also looks red when it should be seen dark. How is this explained? As a handle fluorescence detector was used, the light it emits is possibly green light. Explain all this in greater detail.

Response 13: Thanks for your advice. The observation of Figures 2F and 2G used two different equipment to reveal the expression of DsRed2 in different Developmental stages.

For Figure 2F, according to the reference published by Sun et al, 2018, a laser confocal microscopy (Nikon, Japan) that has a specific excitation wavelength was used to observe the expression of DsRed2 in the different organs of transgenic plants after injection for 50 days. In Figure 2F, due to the specific excitation light, the autofluorescence of transgenic plants not be activated. Therefore, the red fluorescence field of control looks dark.

For Figure 2G, according to the reference published by Huai et al, 2018, a handheld fluorescence observation equipment (LUYOR-3415RG, USA) with green light was used to observe the expression of DsRed2 in the completed transgenic plants in the second year after injection, and the picture was recorded with a camera (Canon, Japan) equid with red filter lens. Therefore, the red fluorescence field of control looks red, and the fluorescence of DsRed2 is expressed as the bright orange.

Comments 14: Lines 145-146: indicate if 82 bulbs were injected, and if 82 plants were obtained from them. You only get one plant per injected bulb? Line 145 is not well written: "showed specific products"??

Response 14: Thanks for your advice. For L. regale, only one plant could be generated from one L. regale bulb. All 82 bulbs we injected generate shoots. In the second year after injection, a total of 4 transgenic L. regale plants with high DsRed2 expression were selected for fluorescence observation by a handheld fluorescence observation equipment. The transgenic plant (T-6) with excellent fluorescence expressed was exhibited in the manuscript. Moreover, the fluorescence of another three transgenic L. regale plants (T-5, T-19, T-21) was observed, the result was shown in Figure 2 of response list. The Line 145 was revised according to the advice.

Figure 2 In the second year after injection, the fluorescence of T-5, T-19, and T-21 observed by handheld fluorescence observation equipment.

4.3 Plant preparation and infection

Similarly, one-year-old L. regale bulbs were collected from the soil before sprouting. A 5 μL bacterial solution was sucked with a syringe, and the needle was vertically inserted into the meristems of L. regale bulbs to inject the A. tumefaciens (Figure 3 and Figure S5). After this injection, each L. regale bulb was cultured at 28°C in the dark for 2 days before being planted in the nutrient soil. In total, 82 L. regale bulbswere subjected to injection.

Comments 15: Figure 4. Same as in figure 2. Figure 4F: Check is the field was green light (see Huai et al (2023) Red fluorescence protein (DsRed2) promotes the screening efficiency in peanut genetic transformation. Front. Plant Sci. 14:1123644. doi: 10.3389/fpls.2023.1123644).

Response 15: Thanks for your advice. According to the reference published by Huai et al, 2018, a handheld fluorescence observation equipment (LUYOR-3415RG, USA) with green light was used to observe the expression of DsRed2. However, the observation of DsRed2 fluorescence must be equid with a red filter lens. Therefore, a camera (Canon, Japan) equipped with red filter lens was used to record the DsRed2 fluorescence. The color of the control plants was red, and the color of fluorescence from DsRed2 expression manifested as the bright orange.

Comments 16: Include the time it takes to obtain transgenic plants of both species with the developed method.

Response 16: Thanks for your advice. The time taken from transformation to obtain transgenic plant was added to the ‘Results’. The content we revised or added was highlighted in red.

2.1. Establishment of Agrobacterium-based injection rhizome transformation method in P. notoginseng

Moreover, Western blot analysis further confirmed the expression of the red fluorescent protein in the leaves of all tested DsRed2 transgenic P. notoginseng plants, including PT-1/3/4/5/8/11/13 (Figure 2E). Additionally, red fluorescence was detected in the leaves, stems, and roots of the DsRed2 transgenic P. notoginseng plant (PT-3) using a laser scanning confocal microscope, but no specific red fluorescence was found in the WT plant, indicating that the red fluorescent protein expressed by DsRed2 in transgenic P. notoginseng possesses normal biological activity (Figure 2F). Furthermore, in the second year after injection, bright orange fluorescence was detected in the rhizome and leaves of DsRed2 transgenic P. notoginseng, but this specific fluorescence was not present in WT plant (Figure 2G). These results demonstrate the successful integration of DsRed2 into the genome of P. notoginseng and the stable expression achieved through the injection of A. tumefaciens into the P. notoginseng rhizome. This method only takes 50-60 days from transgenic operation to the acquisition of positive transgenic plants, and its use means that complicated aseptic operations and tissue culture can be avoided.

2.2. Agrobacterium-based injection in the bulb meristems is a fast and efficient L. regale genetic transformation method

Additionally, Western blot analysis with the DsRed2 antibody revealed that the accumulation of DsRed2 protein in the leaves of positive transgenic L. regale plants (T-6/19/21) was higher than that in the leaves of T-1/5/20. On the contrary, DsRed2 was not found in the leaves of the WT L. regale upon Western blot analysis (Figure 4D). Furthermore, observation with a laser scanning confocal microscope confirmed the presence of red fluorescence in the leaves, stems, roots, and scales of transgenic L. regale plants (T-6), while no specific red fluorescence was observed in the WT L. regale (Figure 4E). Using a handheld fluorescence detector, we found that bright orange fluorescence specifically accumulated in the bulb and shoots of DsRed2 transgenic L. regale plants in the second year after injection (Figure 4F). Generally, through using this method, positive transgenic L. regale can be obtained after 40-50 days. These results, obtained using a method involving Agrobacterium-based injection in the bulb meristems, which proved to be a rapid and efficient approach suitable for the genetic transformation of L. regale, demonstrate the successful integration and stable expression of DsRed2 in transgenic L. regale plants.

  1. Discussion section.

Comments 17: The authors could also discuss the time it takes to obtain a transgenic plant with standard methods and compare it with the times obtained using the developed method.

Response 17: Thanks for your advice. The comparison of time required between the method we established and traditional A. tumefaciens-based transformation methods have been added in the ‘Discussion’. The content we revised or added was highlighted in red.

  1. Discussion

For the Panax spp., there is currently no research indicating that transgenic plants can be successfully developed through the traditionalgenetic transformation method. It is important to note that the transformation method established in this study showed a transformation efficiency of up to 85.7% in P. notoginseng, only taking 50-60 days to achieve this. For Lilium spp., it takes at least 90-120 days to obtain transgenic plants, and the transformation rate is generally only 65.56% [13]. In this study, it only took 40-50 days from the injection of Agrobacterium to the acquisition of transgenic L. regale, and we avoided the complex steps of aseptic operation and tissue culture. At the same time, its efficiency is higher than that of the traditional Agrobacterium-based transformation method, reaching 86.58%, making it a more suitable transformation method in terms of applicability to Lilium spp.

Thanks for your attention and consideration! We appreciate your help very much. Best wishes to you!

Yours sincerely,

Prof. Diqiu Liu (Corresponding author)

Faculty of Life Science and Technology, Kunming University of Science and Technology, Kunming, 650500, China

Reviewer 2 Report

Comments and Suggestions for Authors

Although the manuscript's content describes an Agrobacterium transformation protocol of recalcitrant species to differentiate, the paper is written with low precision. Some terms can not be written in a scientific paper like growth point.  Scientifically the term growth point doesn't exist. The plant or the plant organs grow through meristems (shoot meristems, root meristems,...). Similarly, another concept is dedifferentiation and differentiation. A scientific paper needs rigor. 

Why did the authors choose Lilium sp. and  Panax? it should be clarified because the type of shoot is completely different. Panax has a rhizome and Lilium is a bulb. Would the authors want to compare these species because the meristems are very different). Ginseng has a tap root like carrot and lilium has a bulb. The infiltration process is not clearly described for each of the species. 

The paper is not sufficiently clear.  How do the inserted genes behave in the next propagations in the two species under study?

The paper although interesting has to be subject to a strong revision a report specifically on what happens for each species, why to combine both and what are the main achievements to be used for large-scale propagation and transformation.

Comments on the Quality of English Language

The language must be improved.

Author Response

Dear reviewer,

We have completely revised our manuscript "a fast, efficient, and tissue-culture independent genetic trans-formation method of Panax notoginseng and Lilium regale". The English editing of this manuscript was further revised by MDPI English Editing. And the responses to the Comments are as follows:

Comments 1: Although the manuscript's content describes an Agrobacterium transformation protocol of recalcitrant species to differentiate, the paper is written with low precision. Some terms cannot be written in a scientific paper like growth point. Scientifically the term growth point doesn't exist. The plant or the plant organs grow through meristems (shoot meristems, root meristems,). Similarly, another concept is dedifferentiation and differentiation. A scientific paper needs rigor.

Response 1: Thanks for your advice. We re-checked the terms used in the manuscript. The terms used inappropriately were revised. The content we revised or added was highlighted in red.

Comments 2: Why did the authors choose Lilium sp. and Panax? it should be clarified because the type of shoot is completely different. Panax has a rhizome and Lilium is a bulb. Would the authors want to compare these species because the meristems are very different). Ginseng has a tap root like carrot and lilium has a bulb. The infiltration process is not clearly described for each of the species.

Response 2: Thanks for your advice. The reason we chose Panax spp. and Lilium spp. as experimental models is that they are both perennial plants with low regeneration capacity and low transformation rate using the traditional Agrobacterium-based transformation method. At present, the operation of transformation method used for Panax spp. and Lilium spp. is complicated and takes a long time. In addition, the P.notoginseng and L. regale have meristems in rhizome and bulb, respectively. In spring, the new aboveground part will regenerate from the meristems of the rhizome or bulb, therefore these meristems in P. notoginseng rhizome and L. regale bulb would be the excellent position for genetic transformation. The related contents were added in the ‘Introduction’ of the manuscript. The content we revised or added was highlighted in red.

  1. Introduction

The Agrobacterium-based transgenic technique has been extensively used for gene function validation and genetic modification due to its practicality, cost-effectiveness, and high efficiency [1]. The key steps of this technique include explant preparation, explant infection, co-cultivation with Agrobacterium, explant regeneration, and the screening of transgenic lines [2], but the tissue culture of explant tissue inhibits regeneration and transformation rate [3]. Therefore, the Agrobacterium-based transformation method is particularly difficult to apply to plants with a low regeneration capacity or those with a low transformation rate, such as Panax spp. and Lilium spp. [4]. Panax spp. and Lilium spp.,including Panax notoginseng (Burk) F.H. Chen and Lilium regale Wilson, are both perennial plants with important economic value, but these two species have a low tissue regeneration ability and transformation rate. In addition, the aboveground part of P. notoginseng and L. regalegrows up depending on the meristems located in the rhizome and bulb, respectively. Establishing a fast, efficient, and tissue-culture-independent genetic transformation method is important for the fundamental research and molecular breeding of Panax spp. and Lilium spp. Therefore, the meristems in P. notoginseng rhizome and L. regale bulb would be the excellent position for genetic transformation.

Comments 3: The paper is not sufficiently clear. How do the inserted genes behave in the next propagations in the two species under study?

Response 3: Thanks for your advice. The P. notogiseng and L. regale are perennial plants and have important economic value. It takes 3-5 years and 6-8 years to generate seeds of P. notogiseng and L. regale, respectively. Therefore, it hard to obtain the next generation of transgenic plants at present. How the inserted genes are expressed in the next generation of transgenic plants still needs to research in the future. The related contents were added in the ‘Discussion’ of the manuscript. The content we revised or added was highlighted in red.

  1. Discussion

Recently, a novel Agrobacterium-based transformation method that forgoes the need for sterile operations, developed by Cao et al. [23], has been widely adopted for transgenic plants with high redifferentiation capabilities, such as herbaceous and tuberous plants. However, transformation methods for plants with low differentiation abilities require further exploration. In this study, we introduced a fast and efficient Agrobacterium transformation method based on the injection of A. tumefaciens LBA4404 containing DsRed2 into the meristems of L. regale bulb and P. notoginseng rhizome. This method eliminates the need for sterile operations and meets the requirements for plants with low regeneration abilities. The successful expression of the reporter gene DsRed2 was demonstrated in multiple organs of the P. notoginseng and L. regale plants (Figure 2 and Figure 4). Protein expression of DsRed2 was confirmed via Western blot analysis, and the presence of red fluorescence in multiple organs of positive transgenic P. notoginseng and L. regale plants indicated normal activity of the red fluorescent protein (Figure 2E, F and Figure 4D, E). Most importantly, in the second year after injection, specifically, bright orange fluorescence was observed in the transgenic P. notoginseng and L. regale plants (Figure 2G and Figure 4F), evidencing the stable expression of DsRed2. This study provides robust technical support for fundamental research on plants with low transformation rates and low regeneration abilities. In addition, as perennial plants, it takes 2-3 and 6-8 years for P. notoginseng and L. regale to generate seeds, respectively; therefore, the transgenic plants featured in this study are the chimera plants. Nonetheless, the genetic features of DsRed2 require further investigation in order to clearly understand the stability of the transgenic method established in this study.

Comments 4: The paper although interesting has to be subject to a strong revision a report specifically on what happens for each species, why to combine both and what are the main achievements to be used for large-scale propagation and transformation.

Response 4: Thanks for your advice. The transformation method established in this study successfully produced the transgenic P. notogiseng and L. regale plants that expressed DsRed2 as specific fluorescence. As the economic perennial plants with low regeneration capacity and low transformation rate, the improvement of the transformation methods of P. notogiseng and L. regale provides robust technical support for fundamental research and molecular breeding on plants with low transformation rates and low regeneration abilities. The content we revised or added was highlighted in red.

  1. Introduction

The Agrobacterium-based transgenic technique has been extensively used for gene function validation and genetic modification due to its practicality, cost-effectiveness, and high efficiency [1]. The key steps of this technique include explant preparation, explant infection, co-cultivation with Agrobacterium, explant regeneration, and the screening of transgenic lines [2], but the tissue culture of explant tissue inhibits regeneration and transformation rate [3]. Therefore, the Agrobacterium-based transformation method is particularly difficult to apply to plants with a low regeneration capacity or those with a low transformation rate, such as Panax spp. and Lilium spp. [4]. Panax spp. and Lilium spp.,including Panax notoginseng (Burk) F.H. Chen and Lilium regale Wilson, are both perennial plants with important economic value, but these two species have a low tissue regeneration ability and transformation rate. In addition, the aboveground part of P. notoginseng and L. regalegrows up depending on the meristems located in the rhizome and bulb, respectively. Establishing a fast, efficient, and tissue-culture-independent genetic transformation method is important for the fundamental research and molecular breeding of Panax spp. and Lilium spp. Therefore, the meristems in P. notoginseng rhizome and L. regale bulb would be the excellent position for genetic transformation.

Thanks for your attention and consideration! We appreciate your help very much. Best wishes to you!

Yours sincerely,

Prof. Diqiu Liu (Corresponding author)

Faculty of Life Science and Technology, Kunming University of Science and Technology, Kunming, 650500, China

Reviewer 3 Report

Comments and Suggestions for Authors

In general I think the work is sound and well presented and it describes methods that will make useful contributions to the ease of genetic manipulation of plants that have poor regeneration from tissue culture. In addition to the two species of interest to the authous, I think there are others where these metods will be helpful.

The figures are clear, well-described and infrmative.

Comments on the Quality of English Language

The English language usage is very good but a final edit would be valuable. Just one of several examples that should be changed is there use of the word "secondary" when the intention is surely intended to indicate plants in the second year affter transformation.

Author Response

Dear reviewer,

We have completely revised our manuscript "a fast, efficient, and tissue-culture independent genetic trans-formation method of Panax notoginseng and Lilium regale". The English editing of this manuscript was further revised by MDPI English Editing. And the responses to the Commentss are as follows:

Comments 1: The English language usage is very good but a final edit would be valuable. Just one of several examples that should be changed is their use of the word "secondary" when the intention is surely intended to indicate plants in the second year after transformation.

Response 1: Thanks for your advice, and we checked the use of ‘secondary’ and revised this word. The content we revised was highlighted in red. In addition, the entire manuscript has been language polished by the MDPI Author Services

Thanks for your attention and consideration! We appreciate your help very much. Best wishes to you!

Yours sincerely,

Prof. Diqiu Liu (Corresponding author)

Faculty of Life Science and Technology, Kunming University of Science and Technology, Kunming, 650500, China

Round 2

Reviewer 1 Report

Comments and Suggestions for Authors

Comments and suggestions for authors:

Introduction

The introduction section needs to be improved, it is not easy to read and it lacks fluency and structure. There are some repeated paragraphs and ideas in the text. Below I indicate some suggestions.

Line 38: rewrite the sentences in red by or similar to: “but tissue tissue culture is a limiting step for some species that exhibit a low regenerative capacity and low rate of genetic transformation.”

Lines 39-41: Remove the crossed-out words from the sentence: “Therefore, the Agrobacterium-based transformation method is particularly difficult to apply to plants with a low regeneration capacity or those with a low transformation rate, such as Panax spp. and Lilium spp.”.

Line 41: Remove the crossed out words and the comma after the word Wilson from the sentence:    “Panax spp. and Lilium spp., including Panax notoginseng (Burk) F.H. Chen and Lilium regale Wilson, are both perennial plants with important economic value….”.

Lines 44-49: Move the paragraph and copy it to the end of the introduction as part of the objectives of the work.  Furthermore, replace the underlined words in the sentence: “…the meristems in P. notoginseng rhizome and L. regale bulb would be the excellent position for genetic transformation”. with “could be excellent targets”. 

Line 56: remove “in” in the sentence:  “…. via Agrobacterium-based transformation in [7]”.

Line 59: same as line 56, “in [9]”.

Line 72: remove “In another study,….”.

LIne 75: To increase the fluency of the text, in this sentence “Elsewhere, the transformation rate of Lilium brownie embryogenic callus was found to be 65.56% [15].”, instead of using "elsewhere" you could add the name of the authors of the study, such as:  “Fu and co-authors [15] found ……”.

Line 86: Remove “, in another study,” at the end of the line.

Material and methods

-This section has been considerably improved, but there are still some details to clarify.

-Line 285-286: Was A. tumefaciens injected twice in each position or twice in different positions? It is not clear.

-Line 295: Repace “injetcion” by “injection”.

-Clarifies how the transformation of the rhizome has occurred. When the meristem of the rhizomes is infected with Agrobacterium and these meristems develop, will they give rise to a shoot with a stem and leaves and also roots and a new rhizome?

-The same occurs about the transformation of L. regale bulbs, where it is indicated that roots were taken for analysis. These roots come from the meristem that develops into a complete plant?  

- Another question that is not clear is regarding the second-year plants that are analysed. It should be clarified that in the second year, the plants obtained come from the rhizome that the transgenic plants have developed, and therefore is it transgenic or is the Agrobacterium injected again into the rhizome of the plant?

Results

Explain in the text why the roots are also transgenic. In figure 1 it does not seem that the roots belong to the formed shoot, it seems that it was already in the rhizome. 

Figure 2G, replace red light fluorescence by Green light fluorescence.

The same occurs with the fig 3 about the transformation of L. regale bulbs, where it is indicated that roots were taken for analysis. Draw in the figure the roots emerging from the new shoot.

Figure 4F: replace red light fluorescence by Green light fluorescence.

 Discussion

The discussion needs to be improved; it is only limited to repeating the results obtained. The results obtained could be compared with other works where a transformation method has been developed without the use of tissue culture and indicate similarities and differences between them.

Author Response

Dear reviewer,

We have completely revised our manuscript "A fast, efficient, and tissue-culture independent genetic trans-formation method of Panax notoginseng and Lilium regale". The responses to the comments are as follows:

  1. Introduction

The introduction section needs to be improved, it is not easy to read and it lacks fluency and structure. There are some repeated paragraphs and ideas in the text. Below I indicate some suggestions.

Comments 1: Line 38: rewrite the sentences in red by or similar to: “but tissue tissue culture is a limiting step for some species that exhibit a low regenerative capacity and low rate of genetic transformation.”

Response 1: Thanks for your advice. This sentence was revised to ‘The key steps of this technique include explant preparation, explant infection, co-cultivation with Agrobacterium, explant regeneration, and the screening of transgenic lines [2], but tissue culture is a limiting step for some species that exhibit a low regenerative capacity and low rate of genetic transformation [3].’ The revised content has been marked with red color.

Comments 2: Lines 39-41: Remove the crossed-out words from the sentence: “Therefore, the Agrobacterium-based transformation method is particularly difficult to apply to plants with a low regeneration capacity or those with a low transformation rate, such as Panax spp. and Lilium spp.”.

Response 2: Thanks for your advice. This sentence was revised to ‘Therefore, the Agrobacterium-based transformation method is particularly difficult to apply to plants such as Panax spp. and Lilium spp. [4]’. The revised content has been marked with red color.

Comments 3: Line 41: Remove the crossed out words and the comma after the word Wilson from the sentence: “Panax spp. and Lilium spp., including Panax notoginseng (Burk) F.H. Chen and Lilium regale Wilson, are both perennial plants with important economic value….”.

Response 3: Thanks for your advice. This sentence was revised to ‘For example, Panax notoginseng (Burk) F.H. Chen and Lilium regale Wilson are both perennial plants with important economic value, but these two species have a low tissue regeneration ability and transformation rate.’ The revised content has been marked with red color.

Comments 4: Lines 44-49: Move the paragraph and copy it to the end of the introduction as part of the objectives of the work.  Furthermore, replace the underlined words in the sentence: “…the meristems in P. notoginseng rhizome and L. regale bulb would be the excellent position for genetic transformation”. with “could be excellent targets”. 

Response 4: Thanks for your advice. We removed these sentences to the end of ‘Introduction’, and replaced ‘would be the excellent position’ with ‘could be excellent targets.

Establishing a fast, efficient, and tissue-culture-independent genetic transformation method is important for the fundamental research and molecular breeding of Panax spp. and Lilium spp. However, the transitional Agrobacterium-based transformation method is not suitable for some perennial plants. To address these challenges, we utilized P. notoginseng and L. regale as two experimental models to establish a rapid, efficient, and tissue-culture-independent genetic transformation system. Every spring the aboveground part of P. notoginseng and L. regalegrows up depending on the meristems located in the rhizome and bulb, respectively. Therefore, the meristems in P. notoginseng rhizome and L. regale bulb could be excellent targets for genetic transformation. Additionally, DsRed2 is a red fluorescent protein isolated from Discosoma sp. that was found to be a better reporter gene than GUS and GFP because its excitation light was higher than that of GUS and GFP; thus, it can avoid plant autofluorescence [17]. DsRed2 has been used to screen transgenic cotton (Gossypium herbaceum) and transgenic peanut (Arachis hypogaea) via fluorescence microscopy and handheld fluorescence observation equipment through the detection of red and bright orange fluorescence, respectively [17, 18]. Therefore, Agrobacterium tumefaciens of the LBA4404 strain containing the vector pCAMBIA2300::35S::DsRed2 were injected into the meristems of P. notoginseng rhizome and L. regale bulb. Through 40-60 days of cultivation after Agrobacterium injection, a series of experiments were conducted to evaluate the expression levels and activity of the reporter gene in transgenic plants.

Comments 5: Line 56: remove “in” in the sentence:  “via Agrobacterium-based transformation in [7]”.

Response 5: Thanks for your advice. We removed ‘in’ in the line 56.

Comments 6: Line 59: same as line 56, “in [9]”.

Response 6: Thanks for your advice. We removed ‘in’ in the line 59.

Comments 7: Line 72: remove “In another study,….”.

Response 7: Thanks for your advice. We removed ‘In another study,’ in the line 72.

Comments 8: Line 75: To increase the fluency of the text, in this sentence “Elsewhere, the transformation rate of Lilium brownie embryogenic callus was found to be 65.56% [15].”, instead of using "elsewhere" you could add the name of the authors of the study, such as:  “Fu and co-authors [15] found ……”.

Response 8: Thanks for your advice. We revised this sentence to ‘Fu and co-authors [15] found that the transformation rate of Lilium brownie embryogenic callus was 65.56%’.

Comments 9: Line 86: Remove “, in another study,” at the end of the line.

Response 9: Thanks for your advice. We removed ‘in another study,’ in the line 86.

  1. Material and methods

-This section has been considerably improved, but there are still some details to clarify.

Comments 10: -Line 285-286: Was A. tumefaciens injected twice in each position or twice in different positions? It is not clear.

Response 10: Thanks for your advice. The P. notoginseng rhizome was injected at two positions with once injection per position. The content we added was highlighted in red.

In spring, the soil above the P. notoginseng rhizome was gently removed before the germination of tender shoots. A. tumefaciens LBA4404 containing pCAMBIA2300::35S::DsRed2 was injected into the rhizome of P. notoginseng using a syringe at two positions with once injection per position (Figure S4). A total of 5 μL of bacterial solution was injected into each position (Figure 1). Then, the soil was moved so that it gently covered the rhizome. In total, 71 P. notoginseng rhizomes were injected with A. tumefaciens.

Comments 11: -Line 295: Repace “injetcion” by “injection”.

Response 11: Thanks for your advice. We replaced ‘injetction’ with ‘injection’.

Comments 12: --Clarifies how the transformation of the rhizome has occurred. When the meristem of the rhizomes is infected with Agrobacterium and these meristems develop, will they give rise to a shoot with a stem and leaves and also roots and a new rhizome?

Response 12: Thanks for your advice. After injection with A. tumefaciens, the new stem and leaves are developed from the rhizome. At the same time, some new tuberous roots also are developed from the old taproot, but the P. notoginseng doesn’t generate a new rhizome. The content we added was highlighted in red.

For perennial plants like L. regale and P. notoginseng, the aboveground parts will wither during winter. When the next spring arrives, the tender shoot (stem and leaves) will sprout again from its meristems of rhizome or bulb and grow into a complete plant. At the same time, some new tuberous roots generated from P. notoginseng old taproot, and some new roots and scales also generated form the meristems in L. regalebulb.

Comments 13: -The same occurs about the transformation of L. regale bulbs, where it is indicated that roots were taken for analysis. These roots come from the meristem that develops into a complete plant?

Response 13: Thanks for your advice. After injection with A. tumefaciens, the new stems and leaves were generated from the meristem of blubs. At the same time, some new roots and scales are generated from the meristem of blubs. The content we added was highlighted in red.

For perennial plants like L. regale and P. notoginseng, the aboveground parts will wither during winter. When the next spring arrives, the tender shoot (stem and leaves) will sprout again from its meristems of rhizome or bulb and grow into a complete plant. At the same time, some new tuberous roots generated from P. notoginseng old taproot, and some new roots and scales also generated form the meristems in L. regalebulb.

Comments 14: - Another question that is not clear is regarding the second-year plants that are analysed. It should be clarified that in the second year, the plants obtained come from the rhizome that the transgenic plants have developed, and therefore is it transgenic or is the Agrobacterium injected again into the rhizome of the plant?

Response 14: Thanks for your advice. In the second year after injection, the transgenic plants without additional injection that grew normally and had a high expression of DsRed2 were used for analysis. The content we added was highlighted in red.

4.8. Observing red fluorescence in transgenic plants using a handheld fluorescence detector

In the second year after injection, wild-type plants, as well as transgenic plants without additional injection of A. tumefaciens LBA4404 that grew normally and had a high expression of DsRed2, were collected to observe the bright orange fluorescence derived from DsRed2 expression using handheld fluorescence observation equipment (LUYOR-3415RG, USA), with a green light indicating excitation [18]. Fluorescence observations were recorded with a camera (Canon, Japan) with a red filter lens.

3.Results

Comments 15: Explain in the text why the roots are also transgenic. In figure 1 it does not seem that the roots belong to the formed shoot, it seems that it was already in the rhizome.

Response 15: Thanks for your advice. We improved the description about of transgenic P. notoginseng and revised Figure 1. The content we added was highlighted in red.

4.7. Observation of specific red fluorescence in different organs of DsRed2 transgenic plants using laser confocal microscopy

Three transgenic plants with a high expression of DsRed2 were selected for fluorescence observation. To confirm the biological activity of the red fluorescent protein expressed in the transgenic plants, the leaves, stems, and newly grown tuberous roots of DsRed2 transgenic P. notoginseng plants were collected. Subsequently, they were sliced using a surgical knife. The samples were placed on a glass slide and covered with a cover slip before being observed using a laser confocal microscope (Nikon, Japan) at an excitation wavelength of 560 nm and emission wavelength of 575 nm [17]. Similarly, the leaves, stems, newly grown roots and scales of DsRed2 transgenic L. regale plants were harvested for fluorescence observation using the same method.

Comments 16: Figure 2G, replace red light fluorescence by green light fluorescence.

Response 16: Thanks for your advice. We replaced ‘red light fluorescence’ with ‘green light fluorescence’ in Figure 2G.

Comments 17: The same occurs with the fig 3 about the transformation of L. regale bulbs, where it is indicated that roots were taken for analysis. Draw in the figure the roots emerging from the new shoot.

Response 17: Thanks for your advice. We improved the description of transgenic L. regale and revised Figure 3. The content we added was highlighted in red.

4.7. Observation of specific red fluorescence in different organs of DsRed2 transgenic plants using laser confocal microscopy

Three transgenic plants with a high expression of DsRed2 were selected for fluorescence observation. To confirm the biological activity of the red fluorescent protein expressed in the transgenic plants, the leaves, stems, and newly grown tuberous roots of DsRed2 transgenic P. notoginseng plants were collected. Subsequently, they were sliced using a surgical knife. The samples were placed on a glass slide and covered with a cover slip before being observed using a laser confocal microscope (Nikon, Japan) at an excitation wavelength of 560 nm and emission wavelength of 575 nm [17]. Similarly, the leaves, stems, newly grown roots and scales of DsRed2 transgenic L. regale plants were harvested for fluorescence observation using the same method.

Comments 18: Figure 4F: replace red light fluorescence by green light fluorescence.

Response 18: Thanks for your advice. We replaced ‘red light fluorescence’ with ‘green light fluorescence’ in Figure 4F.

4.Discussion

Comments 19: The discussion needs to be improved; it is only limited to repeating the results obtained. The results obtained could be compared with other works where a transformation method has been developed without the use of tissue culture and indicate similarities and differences between them.

Response 19: Thanks for your advice. We added the content about differences between this method and other tissue-culture-independent transformation methods in the ‘Discussion’ of the manuscript. The content we added was highlighted in red.

Recently, Cao et al. [23] established a novel Agrobacterium-based transformation method named cut-dip-budding that doesn’t need the sterile operations and developed the transgenic herbaceous or tuberous plants. Moreover, this method also was used to generate transgenic succulent plants [24]. The injection A. tumefaciens into the meristems in stems successfully generated some stable transgenic plants, such as the sweet potato (Ipomoea batatas), potato (Solanum tuberosum), and bayhops (Ipomoea pes-caprae) [25]. The above two methods are practicable in those plants which have high redifferentiation capabilities, but the transformation methods for plants with low differentiation abilities need further exploration. In this study, we introduced a fast and efficient Agrobacterium transformation method based on the injection of A. tumefaciens LBA4404 containing DsRed2 into the meristems of L. regale bulb and P. notoginseng rhizome. This method eliminates the need for sterile operations and meets the requirements for plants with low regeneration abilities. The successful expression of the reporter gene DsRed2 was demonstrated in multiple organs of the P. notoginseng and L. regale plants (Figure 2 and Figure 4). Protein expression of DsRed2 was confirmed via Western blot analysis, and the presence of red fluorescence in multiple organs of positive transgenic P. notoginseng and L. regale plants indicated normal activity of the red fluorescent protein (Figure 2E, F and Figure 4D, E).

Yours sincerely,

Prof. Diqiu Liu (Corresponding author)

Faculty of Life Science and Technology, Kunming University of Science and Technology, Kunming, 650500, China

Round 3

Reviewer 1 Report

Comments and Suggestions for Authors

Comments and suggestions for authors:

Line 105. Replace "seedlings" by shoots or plants.

Line 110. Rewrite sentences as or similar to: “leaves from 71 plants derived from injected rhizomes”.

Line 160. Replace “L. regale plants” by “L. regale bulbs”.

Line 161. Rewrite “…and L. regale plants from injected bulbs”.

Line 216. Add “…conventional transgenic plants”

Line 217. Rewrite “.. transformation rate is on average 65.6%”

Author Response

Dear reviewer,

We have completely revised our manuscript "A fast, efficient, and tissue-culture independent genetic trans-formation method of Panax notoginseng and Lilium regale". The responses to the comments are as follows:

Comments 1: Line 105. Replace "seedlings" by shoots or plants.

Response 1: Thanks for your advice. We had replaced ‘seedlings’ by ‘plants’.

Comments 2: Line 110. Rewrite sentences as or similar to: “leaves from 71 plants derived from injected rhizomes”.

Response 2: Thanks for your advice. We rewrite this sentence.

The leaves from 71 plants derived from injected rhizomes were used to extract the genomic DNA, and the wild-type P. notoginseng was included as the negative control in the PCR analysis and subsequent experiments.

Comments 3: Line 160. Replace “L. regale plants” by “L. regale bulbs”.

Response 3: Thanks for your advice. We had replaced ‘L. regale plants’ with ‘L. regale bulbs’.

Comments 4: Line 161. Rewrite “…and L. regale plants from injected bulbs”.

Response 4: Thanks for your advice. We rewrite this sentence.

All L. regale bulbs sprouted new plants from the meristems within 25-30 days after injection, and the L. regale plants from injected bulbs showed the same phenotype as the WT plant (Figure 4A).

Comments 5: Line 216. Add “…conventional transgenic plants”.

Response 5: Thanks for your advice. We added ‘…conventional’ in this sentence.

Comments 6: Line 217. Rewrite “. transformation rate is on average 65.6%”

Response 6: Thanks for your advice. We rewrite this sentence.

It is important to note that the transformation method established in this study showed a transformation efficiency of up to 85.7% in P. notoginseng, only taking 50-60 days to achieve this. For Lilium spp., it takes at least 90-120 days to obtain conventional transgenic plants, and the transformation rate is on average 65.6% [13].

Yours sincerely,

Prof. Diqiu Liu (Corresponding author)

Faculty of Life Science and Technology, Kunming University of Science and Technology, Kunming, 650500, China
